# High Antimicrobial Susceptibility of Cloacal Enterococci and *Escherichia coli* from Free-Living Dalmatian and Great White Pelicans with Detection of Cefotaximase CTX-M-15 Producing *Escherichia coli* ST69

**DOI:** 10.3390/antibiotics14010083

**Published:** 2025-01-14

**Authors:** Teresa Cardona-Cabrera, Sandra Martínez-Álvarez, Carmen González-Azcona, Carlos Javier Gijón-García, Olga Alexandrou, Giorgos Catsadorakis, Panagiotis Azmanis, Carmen Torres, Ursula Höfle

**Affiliations:** 1SaBio Health and Biotechnology Research Group, Institute for Game and Wildlife Research (IREC), Ronda de Toledo 12, 13071 Ciudad Real, Spain; teresa.cardona@uclm.es (T.C.-C.); carlosj94gg@gmail.com (C.J.G.-G.); 2Area of Biochemistry and Molecular Biology, One Health-UR Research Group, University of La Rioja, Calle Madre de Dios 53, 26006 Logroño, Spain; sandra.martinezal@unirioja.es (S.M.-Á.); carmen.gonzalezaz@unirioja.es (C.G.-A.); carmen.torres@unirioja.es (C.T.); 3Society for the Protection of Prespa, Agios Germanos, Prespa, 53150 Florina, Greece; o.alexandrou@spp.gr (O.A.); catsadorakis@spp.gr (G.C.); 4Dubai Falcon Hospital, 22a Street, Zabeel 2, Dubai P.O. Box 23919, United Arab Emirates; azmanis.vet@gmail.com

**Keywords:** Dalmatian pelican, antimicrobial resistance, ESBL-*E. coli*, One Health

## Abstract

**Background/Objectives:** In 2022, an outbreak of H5N1 highly pathogenic avian influenza (HPAI) killed 60% of the largest breeding colony of Dalmatian pelicans (DPs) in the world at Mikri Prespa Lake (Greece), prompting a multidisciplinary study on HPAI and other pathogens. This study determines the antimicrobial resistance rates of cloacal enterococci and *Escherichia coli* in DPs. **Methods:** Fifty-two blood and cloacal swab samples were collected from 31 nestlings (20 DP/11 great white pelicans) hatched after the H5N1 outbreak at the Prespa colony and 21 subadult/adult DPs captured at a spring migration stopover. The swabs were inoculated in non-selective and chromogenic-selective media. Identification was performed using MALDI-TOF, and antimicrobial susceptibility was tested. The genetic content was characterized using PCR and sequencing, and the clonality of extended-spectrum beta-lactamase (ESBL)-producing *Escherichia coli* isolates was characterized using Multilocus Sequence Typing. **Results:** Twenty-eight non-repetitive *E. coli* and 45 enterococci isolates were recovered in non-selective media; most of them were susceptible to all antibiotics tested (85.7% *E. coli*/91.1% enterococci). Three of the fifty-two samples (6%, all adults) contained ESBL-*E. coli* isolates (detected in chromogenic ESBL plates), all carrying the *bla*_CTX-M-15_ gene and belonging to the lineage ST69. **Conclusions:** Despite the susceptibility of most fecal *E. coli* and enterococci isolates to all antibiotics tested, the finding that *E. coli* of lineage ST69 carry *bla*_CTX-M-15_ is of concern. This high-risk clone needs further investigation to elucidate its primary sources and address the growing threat of antimicrobial resistance from an integrated “One Health” perspective. Furthermore, it is imperative to study the potential impacts of ESBL-*E. coli* on the endangered DP further.

## 1. Introduction

The Dalmatian pelican (*Pelecanus crispus*), hereafter referred to as “DP”, is an emblematic Palearctic bird species ranging from Eastern Europe to China [1,2]. Dalmatian Pelicans mainly live in inland freshwater wetlands, rivers, lakes, deltas, and estuaries, where they feed almost exclusively on fish [2,3]. The Southeastern European (SEE) population of DP consists of short-distance migrating populations from Greece, Albania, Montenegro, Bulgaria, Romania, Ukraine, and Turkey [4]. In Greece, the birds belonging to the eastern group of populations (from four breeding colonies: Prespa, Chimaditida, Kerkini, and Karla lakes) winter close to their breeding grounds and usually migrate to wetlands of north/northeastern Greece and western Turkey [4,5] (Figure 1). The largest breeding colony of DPs in the world is located at Mikri Prespa Lake, Greece [6], where they breed side-by-side with the great white pelican *Pelecanus onocrotalus* (GWP) [7], which shows similar feeding behavior and habitat preferences [3]. However, GWPs are long-distance migrants [8] that fly through many important stopover sites in their southward migration to African wetlands along the Rift Valley for wintering in the eastern Africa region [8,9].

Prior to 2022, due to enormous conservation efforts, the SEE DP population was growing, with an increasing population estimated at 2831–3094 breeding pairs [10]. From February to April 2022, an HPAI outbreak (H5N1, clade 2.3.4.4b) resulted in the death of 1734 adult DPs just at Prespa Lake (60% of the colony) and a total of 2286 DPs deceased in Greece, turning this tragic event into the worst ecological disaster for Greek wildlife. Over 40% of the SEE breeding population was lost, corresponding to approximately 10% of the species’ global population. GWPs also died in small numbers, leading to the conclusion that DPs are more susceptible to H5N1 than their congener species [11]. Multidisciplinary research into HPAI and other pathogen carriage in DPs, including bacterial diversity and antimicrobial resistance (AMR) rates, was implemented to investigate the underlying reasons.

AMR has become one of the significant challenges to human and animal health globally [12], and multidrug-resistant (MDR) bacteria, particularly *Escherichia coli,* have been detected in the gastrointestinal tract of wild birds living in a wide range of environments, from very anthropized locations like landfills to remote and less human-influenced ecosystems like the Arctic [13,14,15]. Extended-spectrum beta-lactamase (ESBL)-producing *E. coli* has been designated as a critical priority in the World Health Organization’s list of AMR “priority pathogens”. This is attributed to its resistance to a wide range of antibiotics and its significant clinical relevance, as it is known to cause severe, life-threatening infections [12]. Wild birds harboring ESBL-producing bacteria have been described by several authors in recent years, including other pelican species [15,16,17,18], which puts the spotlight on migratory birds and the role they could play as dispersers due to their ability to connect different ecosystems and remote locations [13,19,20]. It has been reported that DPs visit many different wetlands along their annual cycle [5,21]; thus, even if their potential role as dispersers remains unclear, these birds could be quite useful as sentinels of AMR in the aquatic environment. From a conservation point of view, AMR in bacteria can be linked to strains with pathogenic potential that could negatively impact pelican health.

It will take decades for the DP to recover from the devastating effects of the recent avian influenza outbreak [11]; therefore, determining antibiotic-resistant bacteria that pelicans can carry and excrete is essential to obtain a better understanding of their epidemiology, in addition to highlighting other possible threats for these animals. The present study aims to determine the susceptibility of enteric *E. coli* and *Enterococcus* and the prevalence of ESBL and/or carbapenemase-producing (CP) *E. coli* in pelicans, as well as to understand the role of this species as sentinels of AMR.

## 2. Results

### 2.1. Recovery of E. coli and Enterococci from Cloacal Samples of Pelicans in Non-Supplemented Media

After overnight culture on MacConkey agar (MCA) plates, MALDI-TOF identification, and antibiotic susceptibility testing, a collection of 28 non-repetitive *E. coli* isolates were obtained from 6/21 adults (28.6%) and 20/31 (64.5%) nestlings (10 DPs/10 GWPs). Moreover, after inoculating the cloacal samples in Slanetz–Bartley (S-B) plates, a total of 45 non-repetitive enterococci isolates were obtained from 11/21 adults (52.4%) and 30/31 (96.8%) nestlings (19 DP/11 GWP) (Table 1).

*E. coli* isolates were frequently retrieved from adult DPs (*n* = 8, 28.6% of samples), and out of all the enterococci isolates (*n* = 11, 52.4%), *E. faecalis, E. faecium* (*n* = 4, 19% each), and *E. hirae* (*n* = 3, 14.3%) were the detected species. In the case of nestlings, four enterococcal species were identified, with *E. faecalis* being the most frequent one (*n* = 19, 61.3%), followed by *E. faecium* (*n* = 10, 32.2%), *E. hirae* (*n* = 4, 12.9%), and *E. mundtii* (*n* = 1, 3.2%). *E. coli* was also frequently recovered (*n* = 20, 64.5%). Detailed information about the prevalences and differences found in the nestlings of the two pelican species, as well as the total prevalences for both age groups of DPs, are summarized in Table 1.

### 2.2. Phenotypes and Genotypes of Non-Repetitive E. coli and Enterococcus spp. Isolates Recovered in Non-Supplemented Media

Most of the *E. coli* isolates recovered in MCA media (*n* = 24, 85.7%) were susceptible to all antibiotics tested (Figure 2), with only one isolate from a DP nestling showing resistance to tetracycline (carrying the *tet*(A) gene) and two isolates from adult DPs with resistance to gentamicin. In addition, one isolate from an adult DP showed a multidrug-resistant pattern (MDR, resistant to at least three antimicrobial classes) with phenotypic ESBL production; this isolate carried the *bla*_CTX-M15_ gene, encoding the ESBL CTX-M-15 (Table 2).

Similarly, most enterococci recovered in S-B plates were fully susceptible to all antibiotics tested (*n* = 41, 91.1%) (Figure 3), with four isolates belonging to three adults and one DP nestling showing a resistance pattern to tetracycline (*tet*(M), *n* = 4; *tet*(L), *n* = 2) and streptomycin (*n* = 1) (Table 2). The AMR rates for each antibiotic tested among the *E. coli* and enterococci isolates are detailed in Figure 4.

### 2.3. Prevalence of ESBL-Producing E. coli (ESBL-Ec)

The subsequent seeding of the samples in ESBL- and carbapenem-resistance chromogenic media resulted in the detection of ESBL-Ec isolates in three of the twenty-one adult pelicans (14.3%). As described above, for one of these pelicans, ESBL-Ec was also detected in the non-supplemented MCA media. After the Multilocus Sequence Typing (MLST) of the ESBL-Ec isolates, all of them were assigned to the lineage ST69 (Table 3). None of the nestling samples were positive for ESBL-Ec. None of the pelicans tested (adults or nestlings) showed growth in the carbapenem-chromogenic media, indicating the absence of Enterobacterales with this specific resistant phenotype.

### 2.4. Sexing

From a total of 21 adult pelicans, molecular sexing determined that 12 were males and 9 were females. The DP nestlings were identified as 9 males and 11 females, while the GWPs were identified as 5 males and 6 females.

### 2.5. Differences Among Sex, Age Groups, and Species

A significantly higher prevalence of *E. faecalis* (χ^2^ = 6.304, d.f. = 1, *p* = 0.012) and *E. coli* (χ^2^ = 5.188, d.f. = 1, *p* = 0.023) was found in GWP nestlings, while the occurrence of *E. faecium* was significantly higher in DP nestlings (χ^2^ = 4.188, d.f. = 1, *p* = 0.041) (Table 1). The occurrence of AMR bacteria in nestlings did not differ between the two pelican species. To compare age groups, both nestling species were considered as a whole group without distinction. The prevalence of *E. faecalis* was significantly higher in nestlings (χ^2^ = 9.057, d.f. = 1, *p* = 0.03), and *E. coli* showed a trend of higher prevalence in this age group (χ^2^ = 3.516, d.f. = 1, *p* = 0.061). ESBL-Ec isolates were only found in adults (two males and one female); these were also the only isolates with an MDR phenotype, a finding that was statistically significant (χ^2^ = 4.7, d.f. = 1, *p* = 0.03). Resistant *E. coli* also showed a trend of higher prevalence in adults than in nestlings (χ^2^ = 3.606, d.f. = 1, *p* = 0.058) (Table 1). No differences were found between sexes in adults or nestlings of both species regarding the prevalence of AMR/MDR bacteria, as well as the occurrence of any of the isolated bacterial species.

## 3. Discussion

This is the first general microbiological screening performed for AMR *E. coli* and enterococci in cloacal samples of free-living Dalmatian pelicans; therefore, to the best of our knowledge, all our findings can be potentially considered novel for this animal species. Except for the MDR isolates found in adults (carrying ESBL-Ec), most of the *E. coli* and enterococci recovered were pan-susceptible, showing low rates of resistance to the other antibiotics screened. The differences between the two species of nestlings were scarce, with similar occurrences of AMR and a higher prevalence of *E. coli* in GWPs. The cause or clinical significance of the difference in the predominant species of enterococci has not been elucidated, with *E. faecalis* being more present in GWPs and *E. faecium* more common in DPs. Few significant differences were observed between adults and nestlings, highlighting the ESBL strains mentioned above and the tendency (*p* < 0.1) for *E. coli* with any resistant phenotype to be more prevalent in adults.

As previously mentioned, one important result obtained in this study is the high proportion of pan-susceptible *E. coli* and enterococci isolates (susceptibility to all antibiotics tested) obtained when non-supplemented media were used. These are relevant data that could reflect relatively limited environmental contamination with MDR/AMR bacteria where the animals live. Nevertheless, despite this fact, another remarkable result obtained in this study was the isolation of ESBL-Ec with the *bla*_CTX-M-15_ gene belonging to the ST69 lineage in three adult birds.

Third-generation cephalosporin-resistant *Enterobacteriaceae* are included in the critical group of the WHO Bacterial Priority Pathogen List [12] and, therefore, are considered a highly important finding due to their impact on public health. The importance of these ESBL-Ec isolates lies in the production of enzymes that can hydrolyze third- and fourth-generation cephalosporins and monobactams, in addition to often showing a multi-drug resistance profile, as they combine with other resistance mechanisms as well [22,23]. ESBL-Ec is a significant threat to public health, as therapeutic options are quite restricted, leading to the increased use of carbapenems and consequently increasing the spread of CP bacteria [23]. Among the CTX-M family, CTX-M15 ranks first in prevalence in most parts of the world [23], except Greece, where the SHV type was most frequent [24]. Regarding ST69, it is considered a pandemic lineage like ST131, ST95, or ST73, and it has been frequently reported in both community and nosocomial infections [25,26]. After ST131, this lineage is the most frequent clone, which is present in both humans and animals [26,27,28], and its effective transmission was highlighted in the household transfer of the CTX-M group 9 *E. coli* ST69 strain between pet dogs and humans [29]. The dissemination of this high-risk clone can contribute to an accelerated spread of CTX-M15 beta-lactamase globally and could additionally be spread through avian migration, as was suggested for wintering rooks (*Corvus frugilegus*) [30].

The ESBL exposure of adult pelicans signals the existence of point sources that allow the host–environment circulation of AMR and antibiotic-resistant genes (ARGs). Pelicans primarily use freshwater wetlands, where several antibiotic-resistant bacteria (ARBs) have been described in Greece [31]. ARBs reach rivers and lakes through many sources, such as hospital effluents and wastewater treatment plant discharge, as well as leaching from proximate farms [32,33,34]. It is worth mentioning that antimicrobial consumption in Greece is nearly twofold the average of European Union (EU) countries [35], which explains why the rate of antibiotic resistance in Greece is consistently higher than the EU average [36]. Thus, it could be reasonable to think that these high AMR prevalences could have an impact on the environment through the transfer of ARBs/ARGs to Greek aquatic ecosystems, where pelicans and other wildlife are exposed to them [37].

One important pathway for acquiring ARGs is through livestock, which is considered a significant reservoir of ESBL and has a high prevalence of CTX-M clusters [38,39]. Leachates from farms located near wetlands or lakes can end up in water; even the cattle sometimes come to the lakes to graze and drink and could contribute to AMR/ARG contamination of the aquatic environment [33,34], which can eventually lead to contact with wildlife. Many ESBL-producing *Enterobacteriaceae* have also been reported in a wide diversity of wildlife species [40], although most detections belong to wild birds, with the CTX-M family being the most prevalent. In Greece, ESBL genes in fecal samples from wild birds were first detected in Eurasian magpies (*Pica pica*) [39]. The same research also analyzed samples from livestock, and the results from cattle, pigs, and magpies corroborated the global tendency of all ESBL-Ec harboring *bla*_CTX-M-1/15_. In a more recent survey including 47 different species of wild birds in the same country [18], the *bla*_CTX-M-1_ group (along with *bla*_TEM_) was again isolated in 3.3% of samples. To the best of our knowledge, this is the first report of CTX-M-15 *E. coli* in DPs; the presence of these strains of clinical interest in these pelicans’ guts suggests the need to consider their usefulness as sentinels of AMR contamination in the aquatic environments to which they are closely related.

Finding an epidemic clone like ST69 in these animals is also important due to their spatial ecology. DPs use numerous wetlands around the year [5,21], even visiting five different countries according to previous GPS tracking of the Greek breeding population (Figure 1). In addition, GWPs are long-distance migratory birds that interconnect the European, Asian, and African continents [9]. The migratory movements of both species of pelicans, making use of diverse wetlands across several countries and continents and coming into contact with resident pelicans and other birds, highlight the potential capacity of these animals to carry and disperse AMR and ARGs, as described in other wild birds [14,16,41].

Finally, the ESBL-Ec prevalence found in this study (14.3% of adults sampled) is in accordance with previous reports for other adult wild birds [42,43], although it differs from other studies reporting prevalences higher than 40% [44,45,46]. However, a much lower prevalence was observed in wild birds in Greece [18] and in other European surveys [47,48]. Regional differences in some cases could explain this discrepancy in the results, but another possible explanation suggested by other authors [18] could be the species studied. Some researchers reporting high prevalences have focused on species strongly associated with anthropogenic diets and environments, like seagulls and storks that forage in human waste or landfills [13,48]. A lower prevalence was expected in the nestlings in our study, and no ESBL-Ec was found in these pelicans, although some authors have also described finding it in wild nestlings [15]. Further investigations are needed to look for temporal patterns of ESBL-Ec prevalence and, thus, the persistence of carriage in DPs and GWPs. Expanding sampling to other breeding colonies (including those in which DPs do not come in contact with GWPs) will allow us to obtain more data and elucidate the different factors involved in the acquisition of these critical bacteria in pelicans.

## 4. Material and Methods

### 4.1. Study Sites

Samples were collected from pelicans from two different locations in Greece [Mikri Prespa Lake (N 40°45′, E 21°06′) and Kerkini Lake (N 41°12′, E 23°09′)] (Figure 1); both locations are connected through frequent movements of adult pelicans [21]. Mikri Prespa Lake is the largest breeding colony of DPs in the world, with 1226–1585 pairs located there until 2022 [10], and it is also the nesting site for the majority of the GWPs in Greece, with 500–780 pairs [49]. Both species nest side-by-side, forming large colonies that allow them to strengthen their sense of safety and consequently increase their breeding success [7]. In 2022, after the massive mortality caused by the HPAI outbreak, around 100 pairs of DPs were able to complete the reproductive process and raise 90 chicks [11], while GWPs were not affected. A total of 31 cloacal samples (20 DP/11 GWP) were obtained in July 2022 from nestlings belonging to these remaining breeding pairs.

The second location was Kerkini Lake, which supports the greatest diversity of waterbird species in fresh water in Greece, in addition to being an important wintering and stopover site for many species, including many of the DPs that breed at Prespa [4,5,21]. The HPAI outbreak also affected the DP population in Kerkini, but to a lesser extent than in Mikri Prespa Lake [11]. In January 2023, a total of 21 DPs were sampled at Kerkini Lake, mostly including adults but also some second- and third-year juveniles.

### 4.2. Sampling

Adults were captured using leg-hold traps set on the shore or by hand from a boat after they were attracted to the boat using fish as bait, while nestlings were approached in their nests and gently restrained via manual handling or trapped from a boat using a fishing net pole. Cloacal samples were taken and stored in Amies transport medium (Aptaca, Caneli, Italy), kept refrigerated in the field, and frozen at −20 °C for transport to the laboratory, where they were frozen at −80 °C until further processing. Blood samples were taken from the medial metatarsal vein and kept in sterile tubes with heparin lithium as an anticoagulant. The tubes were spun at 10,000 r.p.m. for two minutes to separate the plasma from the bottom pellet, both of which were kept separated and frozen at −20 °C for further analysis.

### 4.3. Bacterial Isolation and Identification

Cloacal swabs were inoculated in 3 mL of brain heart infusion (BHI) broth for enrichment, and the resulting suspension was incubated at 37 °C for 24 h. For enterobacteria and enterococci isolation, MCA and S-B agar plates were used, adding 50 µL of enriched suspension in each one for streaking over the plates; from these plates, *E. coli* and enterococci were recovered in media not supplemented with antibiotics to obtain a collection of non-selected isolates. After incubation at 37 °C for 24 h, one or two isolated colonies morphologically compatible with *E. coli* (MCA) or enterococci (S-B) were selected and identified via MALDI-TOF mass spectrometry (MALDI Biotyper^®^, Bruker, Vigo, Spain).

Moreover, 50–100 µL of suspension was also spread over chromogenic-selective media for the detection of ESBL-producing bacteria (Brilliance™ ESBL Agar; Oxoid, Thermo Fisher Scientific, Madrid, Spain) and carbapenem-resistant (CP) bacteria (Brilliance™ CRE Medium; Oxoid) and to analyze the prevalence of samples with *E. coli* carrying these specific resistance mechanisms. The plates were incubated at 37 °C for 24 h, and colonies with compatible morphologies were identified via MALDI-TOF, as previously indicated.

### 4.4. Antimicrobial Susceptibility Testing

All isolates identified as *E. coli* or enterococci were tested for antimicrobial susceptibility using the disk-diffusion method, following the methodology and breakpoints of the Clinical Laboratory Standards Institute (CLSI) [50]. Twelve antibiotics were tested for *E. coli*: ampicillin, amoxicillin/clavulanate, cefoxitin, cefotaxime, ceftazidime, imipenem, gentamicin, tobramycin trimethoprim/sulfamethoxazole, chloramphenicol, ciprofloxacin, and tetracycline. The double-disk synergy test, using cefotaxime, ceftazidime, and amoxicillin/clavulanate disks, was used for the detection of phenotypic ESBL production [50]. In the case of enterococci, eleven different antibiotics were tested: vancomycin, teicoplanin, penicillin, ampicillin, tetracycline, erythromycin, gentamicin, streptomycin, linezolid, chloramphenicol, and ciprofloxacin. In the collection of isolates recovered in MCA and S-B plates, non-repetitive isolates were considered for further analysis. These represented one isolate of each species per sample, or more than one if they presented different antibiotic resistance phenotypes. For ESBL-Ec isolates recovered in chromogenic Brilliance™ ESBL Agar media, one isolate per sample was considered and further characterized.

### 4.5. E. coli and Enterococci DNA Extraction

The DNA extraction of *E. coli* isolates was performed by suspending freshly isolated colonies in 1 mL of sterile Milli-Q water and incubating them in boiled water at 100 °C for 8 min. After that, the suspension was thoroughly mixed using a vortex and centrifuged at 12,000× *g* revolutions per minute (rpm) for 2 min. The supernatant with DNA was stored at −20 °C. For the DNA extraction of enterococci isolates, fresh colonies were suspended in 1 mL of sterile Milli-Q water, mixed using a vortex, and centrifuged at 12,000 r.p.m. for 3 min. After removing the supernatant, 20 µL of Insta-Gene matrix (Bio-Rad Laboratories, Hercules, CA, USA) was added to the sediment, mixed using a vortex, and incubated in a bath at 56 °C for 20 min. Another incubation was repeated at 100 °C for 8 min and centrifuged at 12,000 r.p.m. for 3 min. The resultant supernatant with DNA was stored at −20 °C.

### 4.6. Molecular Characterization of Antibiotic-Resistant Genes

Based on the resistant phenotype, genes conferring resistance to tetracycline (*tet*(A)/*tet*(B) for *E. coli* and *tet*(K)/*tet*(M)/*tet*(L) for enterococci), aminoglycosides (*aac*(3′)-*II*), vancomycin (*vanA*/*vanB*), and streptomycin (*str*) were studied using PCR. In addition, all phenotypic ESBL-Ec isolates were selected for *bla*_SHV_, *bla*_CTX-M_, *bla*_TEM,_ and *bla*_OXA_ screening, and the resultant PCR products were further analyzed using sanger sequencing and confirmed by mapping them against the NCBI database. The clonality of ESBL-Ec isolates was characterized using Multilocus Sequence Typing (MLST) based on the Achtman scheme by amplifying and sequencing the seven housekeeping genes (*adk*, *fumC*, *icd*, *gyrB*, *mdh*, *purA* and *recA*) and comparing them with the MLST database to assign sequenced types (ST) (https://pubmlst.org/organisms/escherichia-spp (accessed on 17 November 2023)). The conditions of the PCRs and primers for the tested genes, as well as the molecular typing, are described in Appendix A.

### 4.7. Sex Determination

DNA was extracted from centrifuged blood by taking 5 µL of bottom pellet diluted in 1 mL of Milli-Q water and boiling it at 100 °C for 10 min. The resultant DNA was used in a modified version of the PCR described by Fridolfsson and Ellegren [51], using the commercial kit Phusion blood direct PCR (ThermoFisher, Madrid, Spain). Horizontal electrophoresis in a 1.5% agarose gel was carried out to allow for the separation and posterior visualization of DNA bands.

### 4.8. Statistical Analysis

The data analysis was run in the Statistical Package for Social Sciences (SPSS) Version 28 (IBM, San Francisco, CA, USA). The association between the carriage rate of different bacteria and the bird species/age/sex groups was compared using the chi-squared test at a 95% confidence interval. All analyses were considered statistically significant with probability values less than 0.05.

## 5. Conclusions

The present study describes the prevalence and diversity of enterococci and *E. coli* in the cloacal microbiota of pelican nestlings from two different species, as well as DP adults. Specifically, this study focused on the phenotypic and genotypic characteristics of isolates with clinical interest, with a clonality characterization of ESBL-producing *E. coli* due to its importance in public health.

We can conclude that the high rate of susceptibility of most *E. coli* and enterococci recovered from the tested pelicans is in line with the habitat they live in. Nevertheless, at the same time, the detection of ESBL-producing *E. coli* isolates of the epidemic clone ST69, which carries the gene encoding CTX-M-15, in species like DPs, which have quite restricted diets and moderate influence due to anthropogenic activities, is significant. This finding makes further investigation mandatory to clarify the main sources and address the threat of AMR from a “One Health” perspective.

## Figures and Tables

**Figure 1 antibiotics-14-00083-f001:**
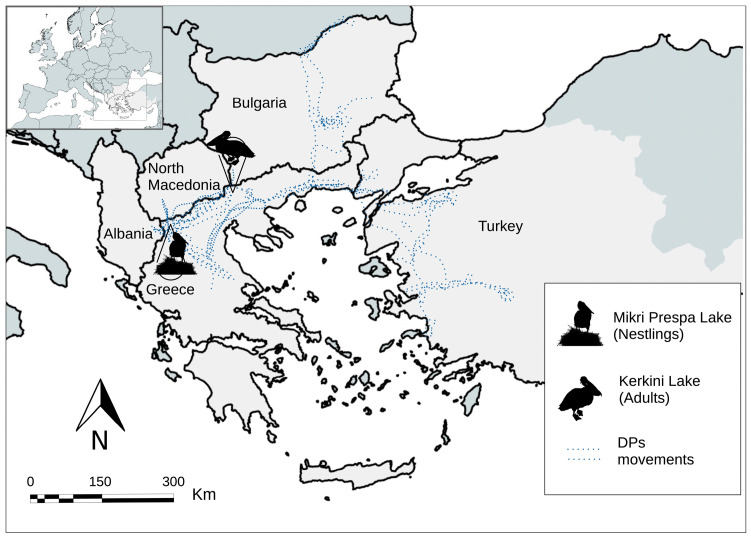
Map showing movements of GPS-tagged DP from the Greek eastern DP population (*n* = 53; 2012–2024). Sampling sites for DP and GWP nestlings (Mikri Prespa Lake) and adult DP (Kerkini Lake) are also shown. Tracking data provided by the Society for the Protection of Prespa. Available online: https://www.movebank.org/cms/movebank-main (accessed on 11 September 2024).

**Figure 2 antibiotics-14-00083-f002:**
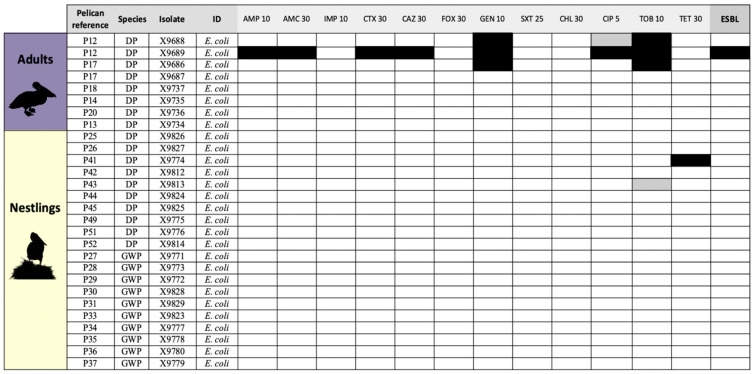
Heatmap showing antibiotic susceptibility (white) and resistance (black) of *E. coli* isolates recovered in MacConkey media from cloacal swabs from great white and Dalmatian pelicans. Legend: DP, Dalmatian pelican; GWP, great white pelican; AMP, ampicillin; AMC, amoxicillin/clavulanate; FOX, cefoxitin; CTX, cefotaxime; CAZ, ceftazidime; IMP, imipenem; GEN, gentamicin; TOB, tobramycin; TET, tetracycline; CHL, chloramphenicol; SXT, trimethoprim/sulfamethoxazole; and CIP, ciprofloxacin. Black squares, antibiotic-resistant phenotype; gray squares, intermediate-resistant phenotype; white squares, wild-type/susceptible phenotype.

**Figure 3 antibiotics-14-00083-f003:**
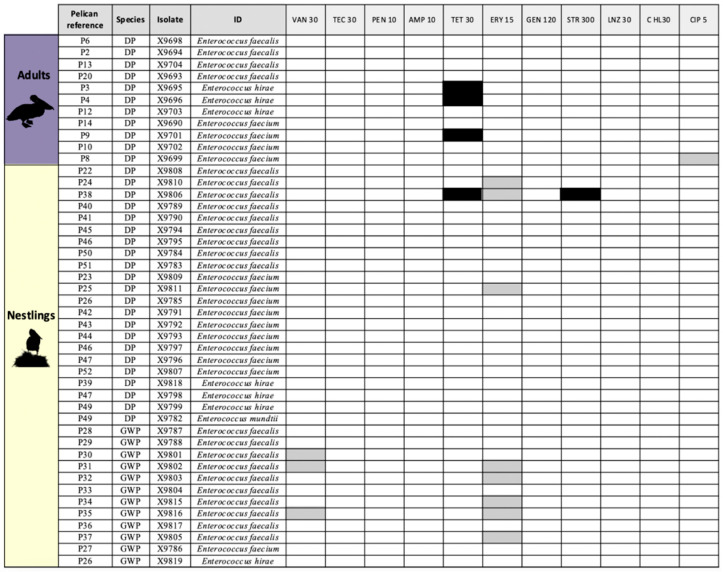
Heatmap showing antibiotic susceptibility (white) and resistance (black) of all *Enterococcus* spp. isolates from great white and Dalmatian pelican cloacal swabs recovered in Slanetz–Bartley media. Legend: DP, Dalmatian pelican; GWP, great white pelican; VAN, vancomycin; TEC, teicoplanin; PEN, penicillin; AMP, ampicillin; TET, tetracycline; ERY, erythromycin; GEN, gentamicin; STR, streptomycin; LNZ, linezolid; CHL, chloramphenicol; and CIP, ciprofloxacin. Black squares, antibiotic-resistant phenotype; gray squares, intermediate-resistant phenotype; white squares, susceptible phenotype.

**Figure 4 antibiotics-14-00083-f004:**
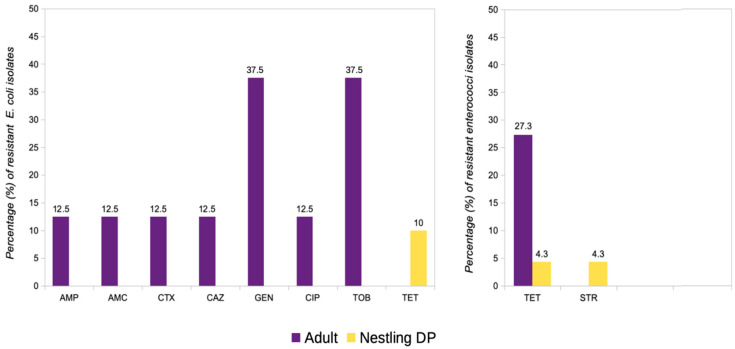
Proportion of isolates exhibiting antimicrobial resistance among cloacal bacteria from adult and nestling Dalmatian pelicans (DP), *E. coli* isolated in MAC (**left**), and enterococci isolated in S-B (**right**). Antibiotics with no resistance detected among the isolates are excluded. Legend: AMP, ampicillin; AMC, amoxicillin/clavulanate; CTX, cefotaxime; CAZ, ceftazidime; GEN, gentamicin; CIP, ciprofloxacin; TOB, tobramycin; TET, tetracycline; and STR, streptomycin.

**Table 1 antibiotics-14-00083-t001:** Detection of *E. coli* and *Enterococcus* spp. isolated in non-selective media (MAC and S-B, respectively) in cloacal samples from adults and nestlings.

Isolate	Number/Percentage of Positive Samples (Number of Non-Repetitive Isolates ^1^)			
	Adults *n* = 21	Nestlings *n* = 31	Nestling DP *n* = 20	Nestling GWP *n* = 11
*Escherichia coli*	6/28.6 (8)	20/64.5 (20)	10/50 (10)	**10/90.9 (10)** ** ***
*Enterococcus faecalis*	4/19 (4)	19/61.3 (19) *	9/45 (9)	**10/90.9 (10)** ** ***
*Enterococcus faecium*	4/19 (4)	10/32.2 (10)	**9/45 (9)** ** ***	1/9.1 (1)
*Enterococcus hirae*	3/14.3 (3)	4/12.9 (4)	4/20 (4)	-
*Enterococcus mundtii*	-	1/3.2 (1)	1/5 (1)	-
Total *Enterococcus* spp.	11/52.4 (11)	30/96.8 (34)	19/95 (23)	11/100 (11)
Total non-repetitive isolates	19	54	33	21

^1^ One isolate of each species and sample or more than one if they presented different resistance phenotypes. Significant differences between age groups are marked as underlined. Significant differences between nestling species are marked in bold. * *p* < 0.05.

**Table 2 antibiotics-14-00083-t002:** Isolates with resistance phenotypes obtained from non-selective media, with detection of resistance genes.

Isolate	ID	Pelican Reference	Age Group	Species	Resistance Phenotype	Resistance Genes
X9689	*E. coli*	P12	A	DP	CTX-AMC-CAZ-GEN-AMP-CIP-TOB	*bla* _CTX-M15_
X9688	*E. coli*	P12	A	DP	GEN-TOB	
X9686	*E. coli*	P17	A	DP	GEN-TOB	
X9774	*E. coli*	P41	N	DP	TET	*tet*(A)
X9695	*E. hirae*	P3	A	DP	TET	*tet*(M)
X9696	*E. hirae*	P4	A	DP	TET	*tet*(M), *tet*(L)
X9701	*E. faecium*	P9	A	DP	TET	*tet*(M), *tet*(L)
X9806	*E. faecalis*	P38	N	DP	STR-TET	*tet*(M)

Legend: A, adult; N, nestling; DP, Dalmatian pelican; GEN, gentamicin; TOB, tobramycin; TET, tetracycline; STR, streptomycin.

**Table 3 antibiotics-14-00083-t003:** Characteristics of ESBL-producing *E. coli* isolates recovered in cloacal swabs of adult DP.

*E. coli* Isolate	Pelican Reference	Resistance Phenotype	ESBL Resistance Genes	ST
X9682	P11	CTX-AMC-CAZ-GEN-AMP-CIP-TOB	*bla* _CTX-M15_	69
X9684 *	P12	CTX-AMC-CAZ-GEN-AMP-CIP-TOB	*bla* _CTX-M15_	69
X9706	P15	CTX-AMC-CAZ-GEN-AMP-SXT-CIP-TOB	*bla* _CTX-M15_	69

* Isolated in both MAC and chromogenic media. Legend: CTX, cefotaxime; AMC, amoxicillin/clavulanate; CAZ, ceftazidime; GEN, gentamicin; AMP, ampicillin; SXT, trimethoprim/sulfamethoxazole; CIP, ciprofloxacin; and TOB, tobramycin.

## Data Availability

Data are reported in the manuscript.

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
