# Peer review of "High Antimicrobial Susceptibility of Cloacal Enterococci and Escherichia coli from Free-Living Dalmatian and Great White Pelicans with Detection of Cefotaximase CTX-M-15 Producing Escherichia coli ST69"

_antibiotics, 2025, doi:10.3390/antibiotics14010083_

Round 1

Reviewer 1 Report

Comments and Suggestions for Authors

The manuscript entitled High susceptibility among cloacal enterococci and Escherichia 2 coli of free-living Dalmatian and great white pelicans, with 3 detection of CTX-M-producing E. coli ST69". First, The study describes the prevalence of enterococci and Escherichia coli in two different species of pelican chicks and adult Dalmatian pelicans. Secondly, The first microbiological screening of free-living Dalmatian pelicans for AMR E. coli and enterococci yielded novel results. It provides the readers with some useful information. It can be considered to publish in Antibiotics, but it needs improvement.

It is recommended that the manuscript be published with minor revisions.

1. Only two sites in Greece were sampled by the authors, and it is hoped that the authors will further justify the sampling of the two sites mentioned above.

2. The authors seem to be inconsistent in the number of birds sampled in nestlings versus adults (Figure 1) and would like to see further clarification from the authors as to the rationale for sampling in this manner.

3. Recommendations about ESBL-producing E. coli can do more narrative.

4. There could be an improvement in essay writing; some sentences are too long making reading difficult, e.g. 335-338 lines are significantly too long.

Author Response

Comments 1: Only two sites in Greece were sampled by the authors, and it is hoped that the authors will further justify the sampling of the two sites mentioned above.

Response 1: Thank you very much for your comment. We agree that it would have been of great interest to have analysed more samples and from more different locations. However as stated in the introduction this study is part of a post-outbreak investigation after the Dalmatian pelican colony in PRESPA had been devastated by an H5N1 outbreak. This study tried to understand the difference in susceptibility of Dalmatian and great white pelicans despite their nesting close together. Thus it took part late in the breeding season. This fact and the fact that capture and sampling of the pelican chicks is extremely labour intensive did not allow for the collection of more samples. In the case of the adults, samples were taken from adults captured for fitting with gps transmitters in order to gain further understanding of movement ecology. Capture of adult pelicans is again labour intensive and very stressful for the birds thus capture of birds for sample collection for this study alone was not justified and we could only sample the number of individuals captured for tagging.

Comments 2: The authors seem to be inconsistent in the number of birds sampled in nestlings versus adults (Figure 1) and would like to see further clarification from the authors as to the rationale for sampling in this manner.

Response 2: Thank You again for this comment, and reasons are partly explained above. Our sampling with view to adults was “opportunistic” in terms that we did not capture animals specifically for the purpose of microbiological analysis, but obtained samples from birds that had been captured for radio-tagging. We could not capture adults at the nesting sites which would have been the ideal thing to do as the stress of handling likely would have affected the chick raising behaviour by the adults and thus the reproductive success in an already affected colony.

Comments 3: Recommendations about ESBL-producing E. coli can do more narrative.

Response 3: Thank You for this recommendation we have now expanded this part of the discussion. You can find the addition in lines 204-220, or below:

The importance of these ESBL-Ec isolates lies in the production of enzymes that can hydrolyze third and fourth-generation cephalosporins and monobactams, besides often showing a multi-drug resistance profile as they combine with other resistance mechanisms as well [22,23]. ESBL-Ec is a significant threat to public health as therapeutic options are quite restricted, leading to increased use of carbapenems and consequently raising the spread of CP bacteria [23].”

Comments 4: There could be an improvement in essay writing; some sentences are too long making reading difficult, e.g. 335-338 lines are significantly too long.

Response 4: Thank You, we have revised the manuscript and shortened sentences that were too long, especially those signalled by the reviewer

Reviewer 2 Report

Comments and Suggestions for Authors

I would like to express my gratitude for offering me the opportunity to review the paper. The article isolated Enterococcus and Escherichia coli carried by Dalmatian pelican(DP), and subsequently determined phenotypic and genotypic characteristics of isolates. Analyzed the impact of DP infection carrying pathogenic bacteria on the surrounding ecological environment, which is meaningful for studying how to alleviate the spread of drug-resistant bacteria.

The experiment is the first comprehensive determination of Escherichia coli and Enterococcus in such a large quantity of DP. Elaborates on the potential impact of DP on the ecological environment during migration from a healthy perspective

For this manuscript, I have the following suggestions:

1. Suggest adding references to relevant research in recent years.

2. The figer clarity is insufficient in the results.

3. Pay attention to writing conventions: such as formatting issues with lines 109, 323, and 351 that need to be corrected

4. Discussion section: The combination of discussion and experimental results is not tight enough. It is suggested to use concise language in lines 215-235.

Author Response

Comments 1: Suggest adding references to relevant research in recent years.

Response 1: Thank you for this recommendation, we have removed some references that were considered not essential and added some new ones. We hope It will be better referenced with these changes.

Comments 2: The figer clarity is insufficient in the results.

Response 2: Thank you for this comment, we have changed the order of some figures for a better narrative line. We have also modified some figures and captions to make them more clear and added some legends in order to make it easier to read and understand.

Comments 3: Pay attention to writing conventions: such as formatting issues with lines 109, 323, and 351 that need to be corrected

Response 3: Thank You again for this comment, we have corrected these issues.

Comments 4: Discussion section: The combination of discussion and experimental results is not tight enough. It is suggested to use concise language in lines 215-235.

Response 4: Thank You for this recommendation, this paragraph has been rewritten and shortened to better focus on point of sources that could offer an explanation of the results obtained. These changes can be seen in lines 230 -240 or you can read it below:

ESBL exposure of adult pelicans signals existence of point sources that allow the host-environment circulation of AMR and antibiotic-resistant genes (ARGs). Pelicans primarily use freshwater wetlands where several antibiotic-resistant bacteria (ARBs) have been described in Greece [31]. ARB reach rivers and lakes through many sources, such as hospital effluents and wastewater treatment plant discharges and leaching from proximate farms [32-34]. It is worth mentioning that antimicrobial consumption in Greece is nearly twofold the average of European Union (EU) countries [35], which explains rates of antibiotic resistance consistently higher than EU average [36]. Thus it could be reasonable to think that these high AMR prevalences could have an impact on the environment through the transfer of ARB/ARGs to Greek aquatic ecosystems, where pelicans and other wildlife are exposed to them [37]”.

Reviewer 3 Report

Comments and Suggestions for Authors

Dear Authors,

Your study provides a valuable contribution to our understanding of antimicrobial resistance (AMR) dynamics in Greek pelican populations and underscores the potential of these birds as important sentinels for AMR surveillance. By documenting the presence of the ST69 Escherichia coli clone within free-living pelican populations, your work offers new insights into the ecology and dissemination of AMR. The detection of CTX-M-producing E. coli further emphasizes the need to examine wildlife as a reservoir of resistance genes.

Your manuscript effectively integrates molecular-level analyses of AMR with ecological and epidemiological perspectives, and the data are presented with scientific rigor, clarity, and precision. This study makes a noteworthy addition to current discussions on AMR and will serve as a valuable reference for future research on the role of wildlife in resistance transmission and evolution.

Only a few minor revisions are needed before publication:

  • Line 84: Replace “Enterococcus” with “Enterococcus spp.
  • Table 1 (Title and last line of the first column): Replace “Enterococcus” with “Enterococcus spp.
  • Figure 1 (Line 132): Italicize “E. coli
  • Figure 2 (Line 140): Replace “Enterococcus” with “Enterococcus spp.
  • Line 249: Correct “CTX-M15” to “CTX-M-15”

I look forward to receiving the revised version of this important work for publication in Antibiotics.

Sincerely,

Author Response

Comments 1: Line 84: Replace “Enterococcus” with “Enterococcus spp.

Response 1: Thank you very much for your suggestion. This issue has been corrected.

Comments 2: Table 1 (Title and last line of the first column): Replace “Enterococcus” with “Enterococcus spp.

Response 2: Thank you for your suggestion. Some aspects on tables have been changed, including your comments.

Comments 3: Figure 1 (Line 132): Italicize “E. coli

Response 3: Thank you for your suggestion. Some aspects on figures have been changed, including this aspect.

Comments 4: Figure 2 (Line 140): Replace “Enterococcus” with “Enterococcus spp.

Response 4: Thank you again for your suggestion. Some aspects on figures have been changed, including this aspect.

Comments 5: Line 249: Correct “CTX-M15” to “CTX-M-15”

Response 5: Thank you for your suggestion. This issue has been corrected.